# Cost-Effectiveness Analysis of the Prophylactic Use of Ertapenem for the Prevention of Surgical Site Infections after Elective Colorectal Surgery

**DOI:** 10.3390/antibiotics10030259

**Published:** 2021-03-04

**Authors:** Costanza Vicentini, Maria Michela Gianino, Alessio Corradi, Noemi Marengo, Valerio Bordino, Silvia Corcione, Francesco Giuseppe De Rosa, Giovanni Fattore, Carla Maria Zotti

**Affiliations:** 1Department of Public Health and Paediatrics, University of Turin, Via Santena 5 Bis, 10126 Turin, Italy; mariola.gianino@unito.it (M.M.G.); alessio.corradi@unito.it (A.C.); noemi.marengo@unito.it (N.M.); valerio.bordino@unito.it (V.B.); carla.zotti@unito.it (C.M.Z.); 2Department of Medical Sciences, Infectious Diseases, University of Turin, Corso Dogliotti 14, 10126 Turin, Italy; silvia.corcione@unito.it (S.C.); francescogiuseppe.derosa@unito.it (F.G.D.R.); 3Department of Social and Political Sciences and CERGAS-SDA, Bocconi University, Via Roentgen 1, 20136 Milan, Italy; giovanni.fattore@unibocconi.it

**Keywords:** antimicrobial resistance, ertapenem, surgical site infections, colorectal surgery

## Abstract

Standard surgical antimicrobial prophylaxis (SAP) regimens are less effective in preventing surgical site infections (SSIs) due to rising antimicrobial resistance (AMR) rates, particularly for patients undergoing colorectal surgery. This study aimed to evaluate whether ertapenem should be a preferred strategy for the prevention of SSIs following elective colorectal surgery compared to three standard SAP regimens: amoxicillin-clavulanate, cefoxitin, and cefazolin plus metronidazole. A cost-effectiveness analysis was conducted using decision tree models. Probabilities of SSIs and AMR-SSIs, costs, and effects (in terms of quality-adjusted life-years) were considered in the assessment of the alternative strategies. Input parameters integrated real data from the Italian surveillance system for SSIs with data from the published literature. A sensitivity analysis was conducted to assess the potential impact of the decreasing efficacy of standard SAP regimens in preventing SSIs. According to our models, ertapenem was the most cost-effective strategy only when compared to amoxicillin-clavulanate, but it did not prove to be superior to cefoxitin and cefazolin plus metronidazole. The sensitivity analysis found ertapenem would be the most cost-effective strategy compared to these agents if their failure rate was more than doubled. The findings of this study suggest ertapenem should not be a preferred strategy for SAP in elective colorectal surgery.

## 1. Introduction

The emergence of antimicrobial resistant (AMR) pathogens and its impact on the treatment of infections is raising concern about the negative effect AMR might have on the efficacy of standard surgical antimicrobial prophylaxis (SAP) regimens in preventing surgical site infections (SSIs), particularly for patients undergoing colorectal surgery [1]. Currently, the recommended agents cover enteric gram negatives and anaerobes, which are the main groups of pathogens associated with SSIs following colorectal surgery [2]. A meta-analysis of 39 randomized controlled trials conducted between 1981 and 2006, in which cefoxitin, cefotetan or cefazolin plus metronidazole were used for SAP, found that the proportion of SSIs after colorectal surgery increased by 5% per year [3]. This trend could be explained by the parallel rise in bowel colonization with AMR bacteria, such as extended-spectrum β-lactamase producing Enterobacteriaceae (ESBL-PE) and *Bacteroides* spp resistant to cefoxitin and cefotetan [4,5,6]. Our prior research suggests the agents recommended by the Italian national SAP guidelines [7] are losing effectiveness in preventing SSIs following elective colorectal surgery [8], further indicating a possible gap in SAP coverage, and highlighting the need for alternative strategies.

Ertapenem, a broad-spectrum carbapenem with activity against ESBL-PE, and superior activity compared to cefoxitin against anaerobes including *Bacteroides* spp. [9], has been approved by the United States (US) Food and Drug Administration and the European Medicines Agency for this indication. In the US, the use of ertapenem for this indication is rapidly spreading: a nationwide study found ertapenem increased from 6% of all prophylactic antibiotics for open colectomies in 2006 to 29% in 2013 [9]. Although ertapenem is effective and widely used for acute abdominal conditions, such as acute appendicitis or diffuse peritonitis in the EU, ertapenem has not been used extensively for the prevention of SSIs after colorectal surgery out of concern that increasing the use of carbapenems will aggravate the levels of resistance to these agents [10].

In 2018, the average carbapenem consumption in the hospital sector in the EU/EEA was 0.04 DDD per 1000 inhabitants per day, which corresponds to around 800,000 carbapenem treatment courses each year [11]. Italy is already facing hyperendemic levels of carbapenem-resistant Enterobacteriaceae [12,13]. According to data from the European Antimicrobial Resistance Surveillance Network (EARS-Net), in 2018 Italy reported the third highest percentage of carbapenem resistance in *Klebsiella pneumoniae* invasive isolates and was the only country that saw a significant increase in the 2015–2018 trend for carbapenem-resistant *Escherichia coli* invasive isolates [10]. Reducing the selective pressure for the development of AMR bacteria and conserving the effectiveness of antibiotics has been recognized as an urgent priority in our country [14].

To date, a limited number of studies evaluating the effectiveness of ertapenem for the prevention of SSIs following elective colorectal surgery compared to different agents or combinations of agents have been published, with mixed results [9,15,16,17,18,19,20,21,22,23]. In the absence of definitive data on the effectiveness and safety of ertapenem for this indication, it is difficult to determine whether ertapenem should be preferred for SAP before elective colorectal surgery, and whether the benefits outweigh the risk of further promoting carbapenem resistance.

Cost-utility analyses using decision tree models allow us to take into consideration the probabilities, costs, and effects in the assessment of alternative strategies. The objective of this study was to evaluate whether ertapenem should be a preferred strategy for the prevention of SSIs following elective colorectal surgery compared to standard SAP regimens.

The primary outcome of this study was to investigate if a prophylaxis with ertapenem was preferable compared to traditional drug protocols, considering the current scientific evidence. The secondary outcome was to define cutoff values of considered variables, at which the strategies became or were no longer cost-effective.

## 2. Results

The input parameters (probabilities, costs, and utilities) and results of the cost-utility analysis are summarized in Table 1. The model template is available as Appendix A.

The outcome was calculated with the shown input parameters and the same WTP.

### 2.1. Probabilities

In 2018, 1626 elective colorectal procedures performed in Piedmont were monitored through Sorveglianza Nazionale delle Infezioni del Sito Chirurgico (SNICh). Patient characteristics are summarized in Table 2.

In total, 119 SSIs and 15 AMR-SSIs were registered. The proportion of SSIs and AMR-SSIs according to each SAP regimen are reported in Table 1.

Figure 1 summarizes the results of the meta-analysis. Full results are provided as Appendix A.

Briefly, both random effects and fixed effects models were computed. Considering the high heterogeneity of the values that were found (I^2^ = 96%, Cochran’s Q *p* < 0.01), the random-effects pooled SSI proportion (0.0952) was used in the cost-effectiveness trees. Only one study included culture and susceptibility testing data [15], which were used to calculate the probability of AMR-SSI for the ertapenem arm of each model.

### 2.2. Costs

Drugs: the costs for SAP spanned from 1.28 € (cefazolin and metronidazole) to 7.79 € (cefoxitin), while the cost of prophylaxis with ertapenem was 39.71 €.

Hospital stay: The cost of a single day of hospital stay was 985 €, but was reduced to 618 € when the operating room and robotic surgery costs were excluded. The mean LOS for procedures with no infection, SSI, and AMR-SSI was 11.2, 20.3, and 35.7 days, respectively. Regarding AMR-SSI, the LOS for procedures for which amoxicillin, cefoxitin, and cefazolin plus metronidazole were used as prophylaxis was 40, 27, and 32 days, respectively. A LOS of 46.4 days was calculated for AMR-SSIs following prophylaxis with ertapenem.

### 2.3. Utility Values

Utility values by day of hospital stay are shown for each outcome in Figure 2. Utility weights at baseline, 7 days, and 30 days of 0.762, 0.514, and 0.714 and 0.718, 0.464, and 0.594, respectively, for the outcomes SSI and no SSI were used to calculate the utility values [24].

The areas under the curve (AUCs) correspond to quality-adjusted life-years (QALYs) for each group. Relevant point coordinates are detailed in Table 1. QALYs were computed up to the 47th postoperative day, in order to maintain consistency with cost calculations, given that the longest LOS included in the models was 46.4 days (AMR-SSIs following prophylaxis with ertapenem).

### 2.4. Cost-Effectiveness Analysis

All models reported dominating strategies: ertapenem dominated amoxicillin-clavulanate, cefoxitin dominated ertapenem, and cefazolin and metronidazole dominated ertapenem. Thus, no incremental cost-effectiveness ratios (ICERs) were calculated. Full results of the analysis are shown in Table 1 and Figure 3a–c.

### 2.5. Sensitivity Analysis

The results of the one-way sensitivity analyses are summarized in Table 3.

Considering the models where ertapenem was not the dominating strategy (i.e., ertapenem vs. cefoxitin and ertapenem vs. cefazolin and metronidazole), a probability of SSI following prophylaxis with cefoxitin or cefazolin and metronidazole of over 8.44% and 8.08%, respectively, would be required for ertapenem to become the dominating strategy.

## 3. Discussion

In this study, we used a decision tree model accounting for both the occurrence of SSI and AMR-SSI to compare the cost-effectiveness of different SAP strategies to prevent SSIs after elective colorectal surgery. According to our findings, in this context ertapenem is a cost-effective strategy only compared to amoxicillin-clavulanate, and it did not prove to be superior to either cefoxitin or cefazolin plus metronidazole. The sensitivity analysis we conducted found ertapenem would be the most cost-effective strategy compared to these agents if their failure rate was more than doubled. Considering a yearly decrease in the effectiveness of both cefoxitin and cefazolin plus metronidazole in preventing SSIs of 5% [3], it would take at least 20 years for ertapenem to become the preferrable strategy according to our model (if the other input parameters remain constant).

In the published literature, there is a lack of cost-effectiveness data supporting the prophylactic use of ertapenem in colorectal surgery. Wilson et al. conducted a post-hoc study on data from patients of the study by Itani et al., comparing per-patient drug and hospitalization costs [15,16]. The study found ertapenem to be cost-saving compared to cefotetan, due to the lower rate of SSI and shorter average LOS in the ertapenem group. However, it should be noted that the superiority of ertapenem compared to cefotetan is probably due to the latter’s limited anaerobic activity [1]. Several more recent studies comparing the effectiveness of ertapenem to standard regimens with effective anaerobic cover do not support the superiority of ertapenem for this indication [9,19,21,22].

A large cohort study conducted in the US, which included both elective and urgent colorectal procedures and was therefore not included in our meta-analysis, found ertapenem to be associated with significantly lower odds of infection compared to cefoxitin, but found cefazolin plus metronidazole to be associated with similar odds of infection to that of ertapenem [21]. According to the results of another nationwide US study, where data from over 90,000 patients undergoing elective and urgent open colectomy in 445 US hospitals were analyzed, ertapenem and cefazolin plus metronidazole also appeared to be associated with a lower risk of infection compared to cefoxitin, although cefazolin plus metronidazole was associated with the lowest odds of infection (44% decreased odds vs. 35% decreased odds in the ertapenem group) [9]. When directly compared to ceftriaxone plus metronidazole, ertapenem did not prove to be more effective in reducing SSIs in a randomized controlled trial conducted in China [19] and was found to be associated with a significantly higher SSI rate in a recent US single-center retrospective study (14% vs. 4.5%) [22]. Accordingly, the 2013 American Society of Health-System Pharmacists guidelines for antimicrobial prophylaxis in surgery recommend a single dose of ceftriaxone plus metronidazole for SAP in colorectal surgery and discourage the routine use of ertapenem for this indication [25]. An option for optimizing SAP and minimizing unnecessary use of carbapenems could be a personalized approach. A targeted approach to SAP, based on preoperative screening for fecal colonization with ESBL-PE, using rectal swab cultures has been proposed for colorectal surgery, similar to rectal swab screening before a prostate biopsy or nasal screening for methicillin-resistant *Staphylococcus aureus* prior to a pacemaker implantation [17]. Results of a recent study suggest patients who are carriers of ESBL-PE have a more than doubled risk of developing SSI after colorectal surgery when standard SAP regimens are used compared to non-carriers, and found that ESBL-PE were more likely to be the causative pathogen among SSIs occurring in carriers compared to non-carriers [26]. ESBL-PE screening prior to elective colorectal surgery and personalized SAP with ertapenem for those who screen positive could be an effective strategy for reducing SSIs in ESBL-PE carriers, according to findings of a recent international quasi-experimental study [17]. The 2016 World Health Organization (WHO) global guidelines for the prevention of SSIs recognized screening and targeted SAP for ESBL-PE carriers as a core topic for SSI prevention in abdominal surgery, although the recommendations were not issued because of a lack of evidence [27]. Diagnostic assays to identify ESBL-PE are time and resource consuming and are therefore currently not widely used in clinical settings [28]. Alternatively, validated algorithms for the prediction of ESBL-PE carriage upon hospital admission developed to guide empiric therapy could be used to inform SAP decisions, especially in high-endemic settings such as Italy and the US [29,30]. These algorithms are based on established predictors of ESBL-PE carriage, such as age, presence of comorbidities, previous antibiotic therapy, recent hospitalization, transfer from another healthcare facility, and recent urinary catheterization. The advantage of these tools lies in their relative simplicity and on the use of patient information, which is usually available upon admission. Because of their high specificity of prediction, these tools could be used by themselves to identify high-risk patients, allowing doctors to limit the use of rectal swabs for a subset of individuals. The effectiveness of ertapenem in preventing SSIs is not only explained by its broad spectrum of activity. Ertapenem has a longer half-life than most standard SAP regimens [1], which eliminates the need for redosing and de-emphasizes the importance of compliance to protocols regarding SAP timing and duration. Antibiotic choice is only one of the many considerations in reducing SSI risk; among other aspects [31], optimizing the administration of standard SAP regimens in terms of timing and duration has also proven to be an effective strategy in this context [8,32]. This study had many limitations that should be considered when interpreting results.

First, there are issues inherent to the study design. Our model is a simplified depiction of management options, outcomes, and other issues related to SAP compared to the real-world situation. A fitting example of this limitation is our evaluation of costs, as we did not consider post-discharge costs, which could be considerable [16], nor did we fully account for the excess costs due to infection with AMR bacteria. A recent study found that the LOS was only the second highest excess cost, due to carbapenem-resistant Gram-negative bacteria in a highly endemic setting, with the highest excess cost attributable mainly to the use of broad-spectrum antibiotics for the treatment of these infections [33]. Therefore, our models could underestimate the true economic impact of AMR-SSI in the ertapenem arms. Further, although it is the first model to consider the economic and clinical burden of both SSI and AMR-SSI in this context, our model did not account for the broader potential effect of the considered SAP regimens on the emergence of patient-specific or community-wide AMR.

Second, our analysis was limited by the availability of data in the published literature, which is extremely lacking in regard to the burden of AMR-SSI. In some cases, we had to resort to the findings of single studies to populate our model, as was the case for the incidence of AMR-SSI following SAP with ertapenem [15], the excess LOS due to carbapenem resistance [34], and the impact of SSI on quality of life [24]. However, we did perform a random-effects meta-analysis to estimate the pooled proportion of SSI after prophylaxis with ertapenem that included over 3000 patients, and we performed broad-range sensitivity analyses to account for the uncertainty of certain variables in our model. This analysis was also limited by the accuracy of our assumptions and the validity of the methodology we applied to extrapolate the input parameters when the data were not available, as in the case of the impact of AMR-SSI on quality of life.

Third, the selection of prophylactic protocols compared to ertapenem was restricted to protocols currently in use in our setting, which excluded some alternatives widely used in the rest of the world. However, our model (Appendix A) can be run with the input parameters of other drug protocols.

For the purpose of this study, our model was populated with real data from our surveillance system, which could limit the external validity of our findings. However, it must be noted that our region employs a highly standardized surveillance system that was established over a decade ago, and that includes a broad range of hospitals in terms of size, ownership, teaching status, and other characteristics; therefore, the results of this study could be relevant in a variety of clinical settings. In any case, our aim was to provide a more comprehensive tool to evaluate the cost-effectiveness of ertapenem for this indication. Evolving AMR patterns that vary not only by region, but often by single hospital, render the evaluation of the real-world effectiveness of the recommended agents critically important. Our data were used here as a case study, but we encourage decision-makers to input their own data in the model (Appendix A) to evaluate whether the use of ertapenem for this indication is justified in their specific setting, and to re-evaluate as pathogens evolve.

In conclusion, despite these limitations, our study suggests ertapenem is not superior to cefoxitin or cefazolin plus metronidazole for the prevention of SSIs following elective colorectal surgery. Even in our highly endemic setting for ESBL-PE, the risks of further promoting AMR and increasing *Clostridium difficile* infection appear not to be justified by the potential benefits of ertapenem, as safer and more cost-effective alternatives are currently available. Our model could be helpful to inform decisions about SAP in different settings, as it allows us to take into consideration the local incidence of AMR pathogens as well as the clinical and financial burden of SSIs.

## 4. Materials and Methods

### 4.1. Study Design and Definitions

Decision tree models were used to compare the cost-effectiveness of ertapenem compared to the three most frequently administered standard SAP regimens for the prevention of SSIs after elective colorectal surgery, as recommended by the national guidelines [7]. The input parameters integrated data from the Italian surveillance system for SSIs (Sorveglianza Nazionale delle Infezioni del Sito Chirurgico, SNICh) [35] with data from the published literature. Three models were constructed: (1) ertapenem vs. amoxicillin-clavulanate, (2) ertapenem vs. cefoxitin, (3) ertapenem vs. cefazolin and metronidazole.

The pathway followed by patients undergoing elective colorectal surgery is summarized in Figure 4.

Patients received either ertapenem or one of the considered standard agents, and transitioned along the arms of the decision tree according to the probability of three outcomes: no infection, SSI, and AMR-SSI, defined as an infection caused by pathogens resistant to the specific prophylactic agent. Every step in the sequence of events leading to an outcome was associated with a cost and utility value expressed in quality-adjusted life-years (QALYs), resulting in a total cost and utility value per strategy based on the likelihood of the sequence of events.

### 4.2. Inclusion/Exclusion Criteria

Procedures were included in compliance with SNICh protocol criteria [35]. All colorectal procedures were considered. An SSI was defined as an infectious event, with an onset in the first 30 days after surgery, with a clear correlation with the surgical procedure.

### 4.3. Probabilities

Standard SAP regimens: The probabilities of SSI and AMR-SSI for the amoxicillin-clavulanate, cefoxitin, and cefazolin plus metronidazole arms of the decision trees were derived from data collected prospectively through SNICh. All monitored elective colorectal procedures performed between 1 January and 31 December 2018, in 42 hospitals of the Piedmont region in the north-west of Italy participating in SNICh, were considered for this analysis.

The methodology for data collection was previously described in detail [8]. Briefly, SNICh is a voluntary, patient-based surveillance system, which employs post-discharge surveillance with a 30 day follow-up period for colorectal procedures. The program is coordinated by public entities (Italian Centre for Disease Prevention and Control, Ministry of Health, Regions of Emilia-Romagna and Piedmont) and, as stated in the protocol [35], the written consent of patients or any authorization from the Ethics Committee is not required.

For each considered prophylactic agent, the number of infections occurring among the procedures in which the agent was administered was retrieved, and the proportion of infections caused by pathogens resistant to the specific prophylactic agent among the infections with available culture data and resistance profiles was applied to determine the probabilities of SSI and AMR-SSI, respectively.

Ertapenem: A systematic review was conducted to determine the efficacy of ertapenem in preventing SSIs, and the proportion of AMR-SSIs following elective colorectal procedures. Further details on the systematic review are provided as supplementary data (Appendix A). Data from all ertapenem study arms were extracted to conduct a quantitative synthesis of the results. Fixed- and random-effects meta-analyses were performed to estimate the overall proportion of SSI. A generalized linear mixed- model meta-analysis method was adopted. The Clopper–Pearson interval method was used to calculate the confidence intervals (CIs) for individual study results.

### 4.4. Costs

Unit costs were valued in euros (2018 prices). The evaluation considered costs from the hospital’s perspective. Because of the short follow-up period, no discounting was applied. The total cost per strategy was calculated as the sum of the drug cost and the cost from the hospital stay. Surgical costs were assumed identical for each outcome (no SSI, SSI, and SSI-AMR), and were therefore excluded from the analysis.

Drug costs: the drug costs were retrieved from the hospital pharmacy of the largest hospital participating in SNICh, the Azienda Ospedaliero Universitaria Città della Salute e della Scienza—Ospedale San Giovanni Battista of Turin, a 1200-bed teaching hospital with primary and secondary referrals, given that the majority of patients were treated in this setting.

Hospital stay costs: The hospital stay costs were calculated by multiplying the average cost of a day of hospital stay by the average number of days of hospitalization (length of stay, LOS) for each outcome. The average cost of a day of hospitalization was calculated on the basis of data from the surgical wards of the San Giovanni Battista hospital. The daily average cost was obtained by dividing the total expenses of surgical wards (excluding the operating room and robotic surgery costs) in the reference year (2018) by the total number of bed days in surgical wards in the same year. The LOS for procedures where no infection or a non-AMR infection was recorded were assumed to be equal for each strategy. For AMR-SSIs, the LOS data derived from SNICh were considered for infections occurring after prophylaxis with the considered standard SAP regimens. The LOS of AMR-SSIs following prophylaxis with ertapenem was calculated as the mean LOS of AMR-SSIs according to the SNICh data, with an addition of 10.7 days to account for carbapenem resistance [33].

### 4.5. Utilities

QALYs were calculated using utility values from a previous study [24]. Specifically, baseline, 7 day-, and 30 day-weights were used to compute utilities for both of the following outcomes: no infection and occurrence of SSI. To the best of our knowledge, no data on utilities pertaining to AMR-SSIs in this context are currently available in the published literature; therefore, these were calculated on the basis of SSI utilities. Specifically, since most of the SSIs in this context occur 7 days after the operation [36], baseline and 7 day-utilities were assumed equal to those of non-AMR SSIs. Further, the utility value at the day of discharge was assumed to be the same in both the SSI and SSI-AMR groups. Given these assumptions, AMR-SSI utilities were calculated. Finally, QALYs were computed as areas under the curve (AUCs).

### 4.6. Statistical Analysis

A cost-utility analysis was conducted, evaluating the outcomes as cost per QALY. Incremental costs and incremental QALYs were calculated as the differences between costs and QALYs between the ertapenem arm and each alternative strategy. The incremental cost-effectiveness ratio (ICER) was calculated when possible (i.e., in non-dominated scenarios). A strategy was defined as cost-effective if the ICER between a strategy and the alternative was below the willingness to pay threshold, set at 40,000 €/QALY. To assess the potential impact of the decreasing efficacy of standard SAP regimens, one-way sensitivity analyses were performed on SSI probability inputs. A threshold value, beyond which a strategy would be dominated by the alternative strategy, was determined for each model, and the decision tree models were created using SilverDecisions [37].

## Figures and Tables

**Figure 1 antibiotics-10-00259-f001:**
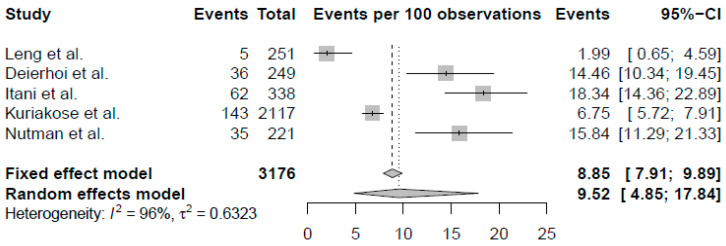
Meta-analysis of the proportion of surgical site infections following prophylaxis with ertapenem in elective colorectal surgery procedures.

**Figure 2 antibiotics-10-00259-f002:**
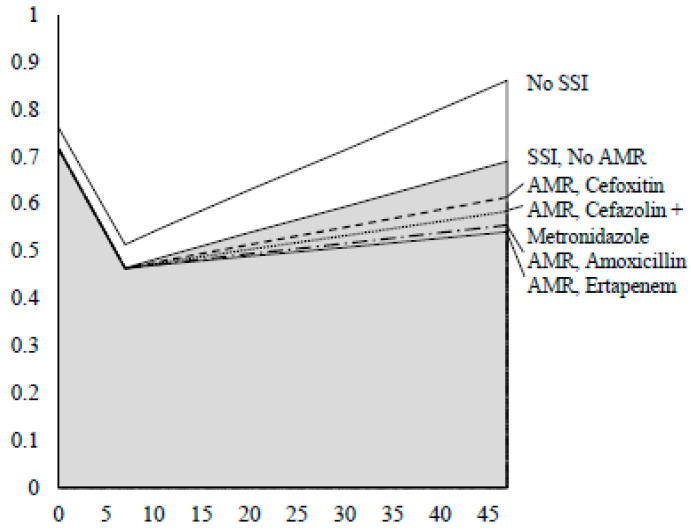
Utility values by day of hospital stay according to the outcomes considered in the cost-effectiveness analysis: no infection, surgical site infection (SSI), and antimicrobial resistant infection (AMR-SSI), per antimicrobial prophylaxis strategy. “SSI”: surgical site infections; “AMR”: antimicrobial resistant microorganisms.

**Figure 3 antibiotics-10-00259-f003:**
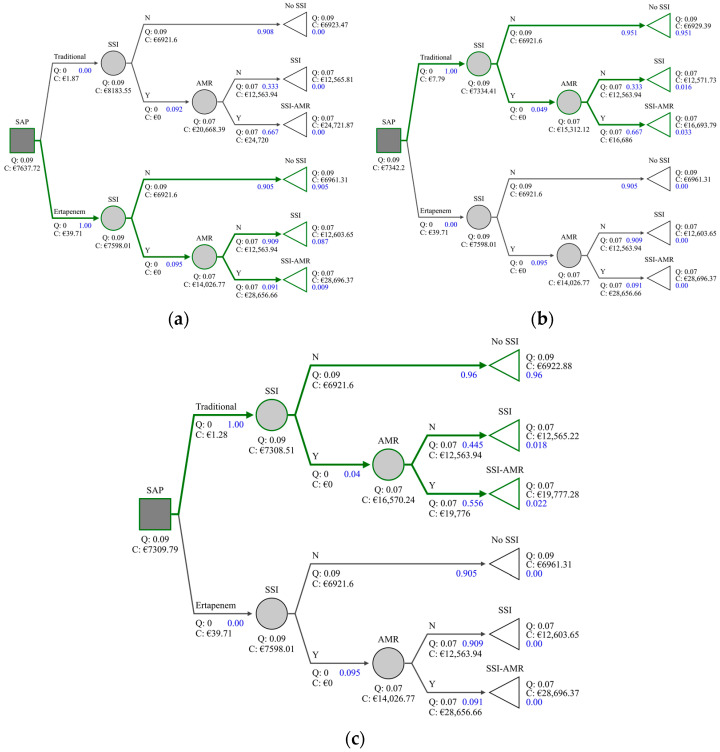
Cost-effectiveness analysis of ertapenem compared to standard prophylactic regimens in elective colorectal surgery. (**a**) Ertapenem vs. amoxicillin-clavulanate; (**b**) Ertapenem vs. cefoxitin; (**c**) Ertapenem vs. cefazolin plus metronidazole.

**Figure 4 antibiotics-10-00259-f004:**
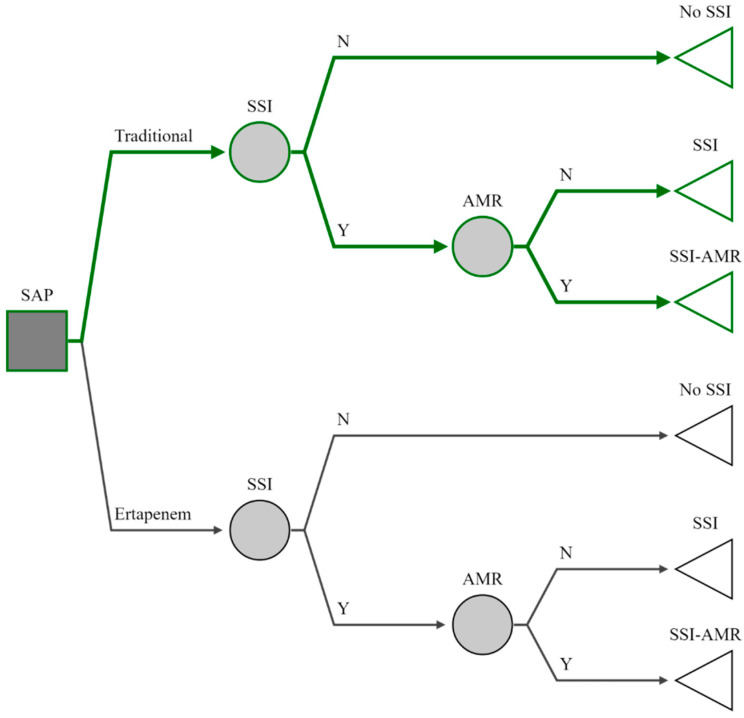
Decision tree model used to evaluate the cost-effectiveness of ertapenem in preventing surgical site infections following elective colorectal surgery compared to standard prophylactic regimens. “SSI”: surgical site infections; “AMR”: antimicrobial resistant microorganisms.

**Table 1 antibiotics-10-00259-t001:** Input parameters and results of the cost-effectiveness analysis comparing ertapenem to standard prophylactic regimens for elective colorectal surgery.

Strategy	Probabilities		QALY			Costs (€)			Outcome ^a^
	SSI	AMR	No SSI	SSI	AMR	No SSI	SSI	AMR	
Ertapenem	9.52%	9.09%	A	B	0.066378	C	D	28,656	-
Amoxicillin-clavulanate	9.18%	66.67%	A	B	0.067188	C	D	24,720	Dominated
Cefoxitin	4.92%	66.67%	A	B	0.070440	C	D	16,686	Dominating
Cefazolin and metronidazole	4.01%	55.55%	A	B	0.068789	C	D	19,776	Dominating
Constant Parameters									
A	0.087623								
B	0.074572								
C	6921								
D	12,564								

^a^ The willingness to pay (WTP) threshold was set to 40,000 €/quality-adjusted life-year (QALY). “SSI”: surgical site infections; “AMR”: antimicrobial resistant microorganisms.

**Table 2 antibiotics-10-00259-t002:** Demographic and clinical characteristics of patients undergoing elective colorectal surgery in Piedmont, 2018.

Characteristic	Value
*Total number of procedures*	*1626*
Patient age, years	
Median (IQR)	71 (62–79)
Female sex, N (%)	909 (55.9)
Infection Risk Index, N (%)	
0–1	1208 (74.29)
2–3	389 (23.92)
Endoscopic or laparoscopic procedure, N (%)	956 (58.79)
Antimicrobial prophylaxis, N (%)	1521 (93.54)
Mean preoperative hospital stay, days	2.4
Mean hospital stay (LOS), days	
No infection	11.2
SSI	22.26
AMR-SSI	35.67
Amoxicillin	40.00
Cefoxitin	27.00
Cefazolin and metronidazole	32.00
Ertapenem	46.37

“SSI”: surgical site infections; “AMR”: antimicrobial resistant microorganisms; “LOS”: length of stay; “IQR”: interquartile range.

**Table 3 antibiotics-10-00259-t003:** Results of the univariate sensitivity analysis comparing ertapenem to standard prophylactic regimens for elective colorectal surgery.

Strategy	Sensitivity Analysis Results		
	Ertapenem Dominated If	Traditional Dominated If	Traditional Cost-Effective If
Ertapenem	-	-	-
Amoxicillin-clavulanate	pSSAP < 5.19% OR pErta > 17.23%	pSSAP > 7.31% OR pErta < 11.96%	pSSAP < 5.30% OR pErta > 16.85%
Cefoxitin	pSSAP < 8.31% OR pErta > 5.63%	pSSAP > 8.44% OR pErta < 5.36%	pSSAP < 8.43% OR pErta > 5.38%
Cefazolin and metronidazole	pSSAP < 7.41% OR pErta > 4.90%	pSSAP > 8.08% OR pErta < 4.72%	pSSAP < 7.45% OR pErta > 4.89%

pSSAP: probability of SSI in standard surgical antimicrobial prophylaxis (SAP) group; pErta: probability of SSI in ertapenem group. The willingness to pay (WTP) threshold was set to 40,000 €/QALY.

## Data Availability

The data presented in this study are available on request from the corresponding author. The data are not publicly available due to privacy restrictions.

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
