# Peer review of "Cost-Effectiveness Analysis of the Prophylactic Use of Ertapenem for the Prevention of Surgical Site Infections after Elective Colorectal Surgery"

_antibiotics, 2021, doi:10.3390/antibiotics10030259_

Round 1

Reviewer 1 Report

The authors evaluated whether ertapenem should be a preferred strategy for the prevention of SSIs following elective colorectal surgery compared to standard SAP regimens. Regarding to the results of their study, the conclusion was that ertapenem should not be a preferred strategy for SAP in elective colorectal surgery.

I read the study with the great interest. First of all I would like to sincerely congratulate the authors on an excellent designed and well written study with a clear message.

I do not have major objections, just few minor remarks and suggestions:

  1. Introduction - I would suggest to the authors to modify sentence in lines 55-57 as follow and to include reference mentioned below: Although ertapenem is effective and widely used for acute abdominal conditions such as acute appendicitis or diffuse peritonitis in the EU, ertapenem has not been used extensively for the prevention of SSIs after colorectal surgery out of concern that increasing the use of carbapenems will aggravate the levels of resistance to these agents. REFERENCE: Pogorelić Z, Silov N, Jukić M, Elezović Baloević S, Poklepović Peričić T, Jerončić A. Ertapenem Monotherapy versus Gentamicin Plus Metronidazole for Perforated Appendicitis in Pediatric Patients. Surg Infect (Larchmt). 2019;20(8):625-630. doi: 10.1089/sur.2019.025.
  2. Tables - All abbreviations used in Tables should be mentioned in legend below the Table. Each entirely Table should be on one page, not half on one and half on the other. Also the titles of the Tables and Figures should be on the same page as Table / Figure.
  3. Methodology – Please provide clear inclusion / exclusion criteria.
  4. Methodology – Please provide primary and secondary outcomes of the study.
  5. Was the study approved by the ethics committee? Regardless of whether it is a retrospective study, the study should be approved by the ethics committee. Please provide IRB reference.

Author Response

27/02/2021

Dear Editors,

We are submitting the revised version of our manuscript “Cost-Effectiveness Analysis of the Prophylactic Use of Ertapenem for the Prevention of Surgical Site Infections after Elective Colorectal Surgery”. We would like to thank the Editors and the expert Reviewers for their time and for their insightful comments and suggestions. We hope to have sufficiently improved on the issues present in our original manuscript. We are extremely thankful for the opportunity to better define and expand on some aspects of our research.

Reviewer #1:

  1. Introduction - I would suggest to the authors to modify sentence in lines 55-57 as follow and to include reference mentioned below: Although ertapenem is effective and widely used for acute abdominal conditions such as acute appendicitis or diffuse peritonitis in the EU, ertapenem has not been used extensively for the prevention of SSIs after colorectal surgery out of concern that increasing the use of carbapenems will aggravate the levels of resistance to these agents. REFERENCE: Pogorelić Z, Silov N, Jukić M, Elezović Baloević S, Poklepović Peričić T, Jerončić A. Ertapenem Monotherapy versus Gentamicin Plus Metronidazole for Perforated Appendicitis in Pediatric Patients. Surg Infect (Larchmt). 2019;20(8):625-630. doi: 10.1089/sur.2019.025.

The sentence was modified accordingly. We thank the Reviewer for the recommendation, we have included the suggested reference in our paper.

  1. Tables - All abbreviations used in Tables should be mentioned in legend below the Table. Each entirely Table should be on one page, not half on one and half on the other. Also the titles of the Tables and Figures should be on the same page as Table / Figure.

We adjusted all the tables in the correct format, as the Reviewer kindly suggested. However, we kindly ask the Editors to further modify the manuscript in its pre-proof version, if necessary.

  1. Methodology – Please provide clear inclusion / exclusion criteria.

We would like to thank the reviewer for the opportunity to expand on this subject.

We added the following paragraph in the Materials and Methods section (lines 360-363):

Inclusion/exclusion criteria

Procedures were included in compliance with SNICh protocol criteria [35]. All colorectal procedures were considered. An SSI was defined as an infectious event, with an onset in the first thirty days after surgery, with a clear correlation with the surgical procedure.

  1. Methodology – Please provide primary and secondary outcomes of the study.

We added the following sentences at the end of Introduction paragraph (lines 88-91):

The primary outcome of this study was to investigate if a prophylaxis with ertapenem was preferable compared to traditional drug protocols, considering the current scientific evidence. The secondary outcome was to define cutoff values of considered variables at which the strategies became/were no longer cost-effective.

  1. Was the study approved by the ethics committee? Regardless of whether it is a retrospective study, the study should be approved by the ethics committee. Please provide IRB reference.
We reported in the paragraph Institutional Review Board and Informed Consent Statement  that: “As stated in the SNICh protocol [35], considering the programme’s aims are disease surveillance and healthcare quality improvement, and that the programme is coordinated by public entities (Italian Centre for Disease Control, CCM, Ministry of Health, Regions of Emilia-Romagna and Piedmont), the written consent of patients involved in surveillance or any other authorization from the Ethics Committee and/or the Protection Commissioner is not requested.”

Reviewer 2 Report

The authors present an analysis on the cost-effectiveness of ertapenem as a surgical prophylaxis agent. This is a timely article as hospitals have been moving toward agents other than cefoxitin and cefotetan for colorectal surgical prophylaxis. Even within our hospital system some ministries have switched to ertapenem for these cases. 

Is there a specific reason ceftriaxone + metronidazole was not explored for cost-benefit, other than a previous study looking into it? While I do not feel it is necessary for the authors to go back and include it, it would have been interesting to include this. The authors should state why it was not included within the manuscript as readers would likely wonder since it is an option for colorectal surgery and ceftriaxone being once daily dosing and metronidazole with its longer re-dose schedule (~8h) also make the regimen appealing. 

A sentence in the introduction is missing "the": "Our prior research suggests agents recommended by the Italian national SAP guidelines"

The sentence "Ertapenem, a broad-spectrum carbapenem with activity against ESBL-PE and superior activity against anaerobes including Bacteroides spp" should clarify "superior" compared to what? One would argue that metronidazole has superior activity compared to ertapenem, particularly when looking at B fragilis and B ovatus.

The authors state anaerobe breakthrough infections becoming increasingly common with cefotetan, they should also mention cefoxitin also. I believe there is published literature showing this. This certainly has occurred at our institution, leading to the switch from cefoxitin to cefazolin + metronidazole.

The author can consider the addition of the following article in support of carbapenems leading to resistance in Gram-negative bacteria:

Journal of Hospital Infection 2011 May;78(1):54-8. doi:10.1016/j.jhin.2011.01.014

ICHE 2010 Dec;31(12):1242-9. doi:10.1086/657138. Epub 2010 Oct 28.

Author Response

27/02/2021

Dear Editors,

We are submitting the revised version of our manuscript “Cost-Effectiveness Analysis of the Prophylactic Use of Ertapenem for the Prevention of Surgical Site Infections after Elective Colorectal Surgery”. We would like to thank the Editors and the expert Reviewers for their time and for their insightful comments and suggestions. We hope to have sufficiently improved on the issues present in our original manuscript. We are extremely thankful for the opportunity to better define and expand on some aspects of our research.

Reviewer #2:

  1. Is there a specific reason ceftriaxone + metronidazole was not explored for cost-benefit, other than a previous study looking into it? While I do not feel it is necessary for the authors to go back and include it, it would have been interesting to include this. The authors should state why it was not included within the manuscript as readers would likely wonder since it is an option for colorectal surgery and ceftriaxone being once daily dosing and metronidazole with its longer re-dose schedule (~8h) also make the regimen appealing.

We thank the Reviewer for sharing with us his/her interesting clinical experience and for suggesting another possible application of our analysis. We did not explore the cost-benefit for ceftriaxone+metronidazole because, as we explained in the Materials and Methods section, “Decision tree models were used to compare the cost-effectiveness of ertapenem compared to the three most frequently administered standard SAP regimens for the prevention of SSIs after elective colorectal surgery recommended by national guidelines”. However, we agree with the Reviewer that this analysis would be of great interest for the reader. For this reason, we added the following sentence in the Discussion section, as a limitation (lines 300-303):

Third, the selection of prophylactic protocols compared to ertapenem was restricted to protocols currently in use in our setting, which excluded some alternatives widely used in the rest of the world. However, our model (Supplementary File 2) can be run with input parameters of other drug protocols.

  1. A sentence in the introduction is missing "the": "Our prior research suggests agents recommended by the Italian national SAP guidelines"

We have corrected the mistake. Thank you.

  1. The sentence "Ertapenem, a broad-spectrum carbapenem with activity against ESBL-PE and superior activity against anaerobes including Bacteroides spp" should clarify "superior" compared to what? One would argue that metronidazole has superior activity compared to ertapenem, particularly when looking at B fragilis and B ovatus.

We have clarified the sentence: superior comparing to cefoxitin (line 50).

  1. The authors state anaerobe breakthrough infections becoming increasingly common with cefotetan, they should also mention cefoxitin also. I believe there is published literature showing this. This certainly has occurred at our institution, leading to the switch from cefoxitin to cefazolin + metronidazole.

We thank the reviewer for this suggestion. We included a new reference (ref. 6) in the Introduction section (line 45).

  1. The author can consider the addition of the following article in support of carbapenems leading to resistance in Gram-negative bacteria: Journal of Hospital Infection 2011 May;78(1):54-8. doi:10.1016/j.jhin.2011.01.014

We added the suggested reference that supports our consideration of carbapenems leading to resistance in Gram-negative bacteria (lines 68-69). We thank the Reviewer for recommending this interesting study.

Again, thank you for your time and consideration.